# Strategies for Using ICT Skills in Educational Systems for Sustainable Youth Employability in South Africa

Abiodun Alao *[ID] and Roelien Brink

Applied Information Systems Department, School of Consumer Intelligence, and Information Systems, College of Business and Economics, University of Johannesburg, Johannesburg 2028, South Africa
* Correspondence: alaoa@uj.ac.za

**Abstract:** Information and Communication Technology (ICT) can play a significant role in the socioeconomic development of many countries. Digitisation in South Africa has increased, and ICT skills are pivotal in the sustainability of youth employability in the labour market. Hence, ICT skills, soft, hard, and technical skills are required in government, private organisations, and businesses. This study aims to investigate possible ways educational systems can adopt ICT skills to improve youth employability in South Africa. This study examines the factors that affect youth employability such as lack of ICT skills, access, income, affordability, infrastructure, poverty gap, inequality, lack of education, lack of access to information, and high demand for IT skills expectations in organisations as among the challenges that hinder youth employability in the South African economy. We propose that educational institutions should incorporate practical pedagogy to prepare qualified youths for the labour market. This study focuses on using ICTs for the sustainable development of youth employability in South Africa. The Sustainable Livelihood Theory was used as the study framework while the quantitative method was used for the data collection process. The researchers used close-ended and open-ended questions to draft a questionnaire to gather data from 49 respondents. We triangulated the received data from youths living in the East Rand of Johannesburg. Results derived from the study show the significance of ICT skills in educational systems on youth employability. The practical implication of the study recommends that policymakers implement ICT skill strategies to support educational institutions to prepare youths for the labour market.

**Keywords:** educational systems; ICT devices; ICT skills; youth employability; labour market

## 1. Introduction

There has been a continuous change in many organisational sectors globally due to the Fourth Industrial Revolution (4IR) or Industry 4.0. The 41R is the 21st-century rapid change to technology in industries and societal pattern processes. This has increased inter-connectivity and smart automation in organisational operations [1]. The integration of the Fourth Industrial Revolution (4IR) and the Third Industrial Revolution (3IR) promotes advanced digital and ICT skills which consist of electronic technologies and industrial techniques used to administer information to improve computerisation [2]. The goal of both revolutions is to develop the industry and economy of any country which makes their use relevant [1,3]. The 4IR performs the function of computers and computational thinking (CT) and mechanisms of artificial intelligence (AI) and other advanced technologies that have become relevant in many sectors [2]. Hence, digitisation is essential to the management of organisations such as businesses and health, manufacturing, agriculture, construction, education, and many other sectors [4].

The South African government has realised the importance of ICT skills development and wants to adopt suitable ICT measures that can contribute to the growth of the country's economy and youth employability. This study emphasises that digitisation should be integrated into pedagogy to train prospective youths for the labour market. This is because

the supply and demand for intermediate-level ICT skills in South Africa are limited, causing high youth unemployment in the country. Therefore, educational systems need to strategize to adopt practical measures that can promote ICT skills among youths in South Africa. In addition, work-integrated learning methods that focus on impacting ICT projects that relate to real-life practical industry experience prepare youths for the many industry needs. Especially since industries have incorporated technological strategies to improve their daily operational activities, ICT skills have become imperative for the sustainability of youth employability in the labour market.

Sustainability is defined as meeting the needs of people without compromising the capabilities of future generations [5]. The sustainability of youth employability can be improved through the use of ICT skills, that is, the ability to operate and have an in-depth knowledge of a range of technology software using digital devices such as mobile devices: phones, tablets, and laptops [5]. Likewise, digital skills are defined as the ability to evaluate, share, discover and create content using digital devices such as smartphones, laptops, computers, and others. ICT or digital skills can allow youths to access information, communication applications, and internet networks relevant to managing information received online and other specialised programs which can be used for their personal development [5]. For this inherent purpose, this study examines how ICT skills adoption can improve youth employability in South Africa and poses the research question: How can educational systems adopt ICT skills to improve youth employability in South Africa?

Study shows 85.7% of unemployed youth candidates are applying for jobs in the labour market without suitable information and communication technology (ICT) qualifications or ICT skills relevant to the labour market [6]. Hence, the rate of unemployment in South Africa increased to 55.75% in 2020, and youth unemployment increased to 66.5% in 2021 among 15–24 years old but decreased to 64% in 2022 [7–9]. The deficient demand for labour and increased skill-intensive orientation for the South African economy have caused the substandard supply to the emergence of risky and low ICT skills among youths in the country [10]. Therefore, education systems can develop practical measures that will allow learners to familiarise themselves with technologies that employees use in industries to prepare prospective youths for the labour market.

History has shown that from 1994 to 2019, 10% of South Africa's richest population has become richer, and the poor have become poorer because the percentage of total income has reduced significantly [10,11]. This has affected the South African economy, and youth employment is relatively low. However, this can be rectified through the incorporation of ICT skills development training in vocational schools and education institutions [12]. Although access to technology can be challenging due to the lack of technology access, infrastructure, and resources in South African society causing the lack of ICT skills among many youths at the grassroots level [13].

*Research Problem*

The researchers identified the research problem of this study which focused on the lack of training and adoption of advanced ICT skills in the pedagogy of educational systems, especially as the efforts of the South African government have not thoroughly addressed issues focused on youth unemployment in the country. In addition, we explored the factors that contribute to youth unemployment such as lack of ICT skills, access, affordability, infrastructure, poverty gap, inequality, lack of education, lack of information access, and high demand for IT skills expectations in organisations [14]. For this purpose, this study addressed an alternative solution to promote ICT skills training at the grassroots level using government-sponsored public technology access points to prepare youths for industry expectations [15]. Furthermore, the study adopted the Sustainable Livelihood Theory to explain how youths can use ICT skills to improve their livelihood.

## 2. Study Context

Many organisations require workers to be knowledgeable about ICT skills and direct interaction with technology. Countries such as South Africa have a high unemployment rate and require trained ICT professionals to boost the country's economic development and limit youth unemployment. Hence, it is important to highlight that there is a shortage of youth professionals in South Africa due to the lack of industry skills, such as advanced ICT skills, that hinder youth employability [16]. Other challenges that affect youths in the country include poverty, inequality, skills mismatch, and high skills expectations [14].

Studies have shown that relevant ICT skill gaps in the labour market are yet to be addressed to reduce the country's economic meltdown of youth unemployment [5,17]. ICT skills are work capabilities that youths need to gain to operate a wide range of technology software. Youths need to understand the organisational preferences of technology daily duties to have an idea of the operational management of industries, especially since many organisations require their employees to be equipped with ICT skills. These relevant skill developments include communication skills, customer service, scheduling and time management, project management, analytical thinking, flexibility, the ability to work independently, and technical skills. These are essential soft skills in demand in many businesses, industries, and private-owned entrepreneurship [5]. These skills are significant in the workplace for hybrid, remote or full-time industry employees.

Therefore, education systems will need to strategize ways to incorporate practical techniques into their teaching curriculum that can prepare learners for organisational work. Hence, youth unemployment can be improved through enhanced information technology (IT), advanced ICT skills, and industrial and practical work experience through the implementation of work-integrated learning at educational institutions for youths to have a competitive advantage in the labour market.

### 2.1. ICT Skills Required for the Labor Market

Recently, digitisation changed the way organisations perceive the importance of ICT skills in the labour market [4]. Especially as ICT careers are the most crucial professions around the world, the high demand for ICT professionals in the labour markets has shown that there is a shortage of skilled ICT professionals in South Africa [2,18]. Information and Communication Technology careers include software development, information technology (IT) audit, IT infrastructure and architecture, IT change management, database development, system design, business analysis, ICT network support and other technical skills in high demand in organisations [2,19].

Studies show employers have identified the required skills in the recruitment process, which was caused by a shortage of qualified employees with advanced digital skills [20,21]. As a result, employers emphasise that a lack of soft skills, such as communication, project management, negotiation, and leadership skills are significant components that education systems need to incorporate into their curriculum [2]. Extensive studies described that ICT skills are required in many technical fields, and soft skills are mostly significantly needed in industries. Hence, to build the confidence to have digital skills ability, youths need to have practical access to digital tools at vocational, technical, and educational institutions.

Many youths lack ICT, soft, hard, and technical skills relevant to the labour market which is causing a gap that is yet to be bridged, as many youths did not have the opportunity to have the practical aspect of using advanced ICT tools while studying. The technical knowledge and advanced ICT skills that align with the organisation's operations are a prerequisite for prospective employees. Therefore, educational institutions need to strategize to implement adequate practical teaching techniques into the South African school system [2]. The National Workforce Centre for Emerging Technologies described information technology (IT) as relevant to the labour market [22]. The sectors classified information communication technology skills as a high priority in any career and categorised different skills into a pyramid of relevant ICT skills required in industries such as private organisations, government, and educational institutions [22]. The pyramid categorises ICT

skills into three tiers. In addition, the ICT skills pyramid is used to explain how soft, hard, and technical critical skills are relevant in organisational operations [22]. The study shows different ICT pyramids categorised into three tiers to explain the relevance of ICT skills in the labour market (see Figure 1).

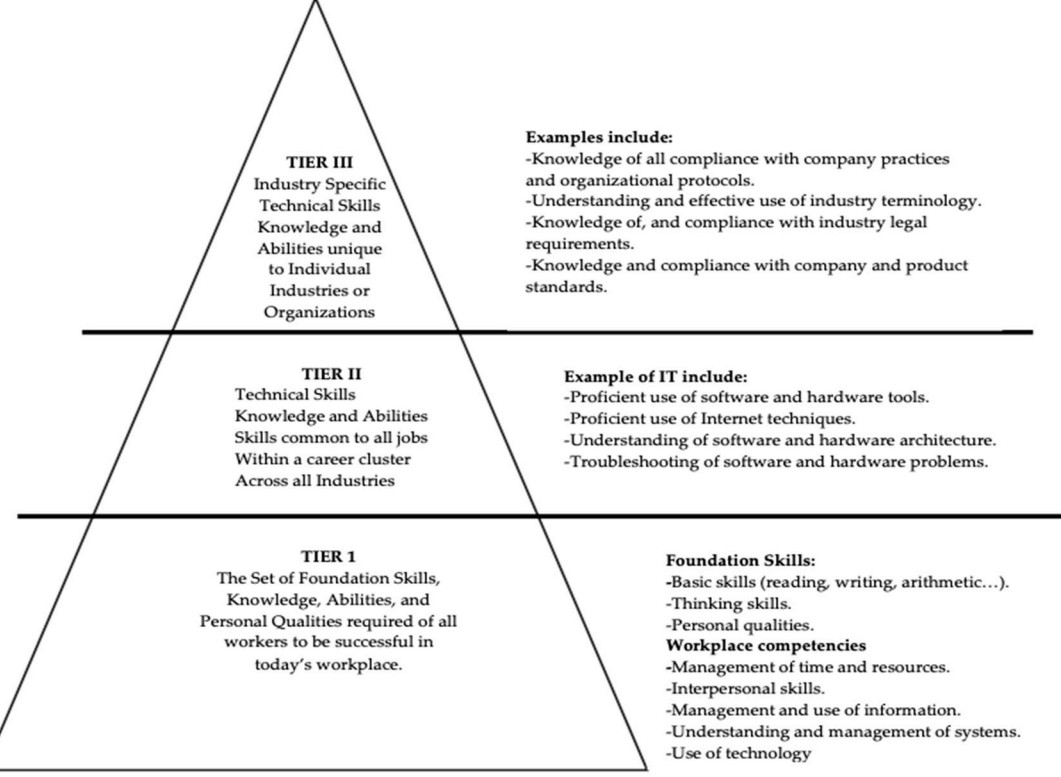

**Figure 1.** IT Skill Pyramid [22].

Furthermore, the study shows the importance of ICT skills to improve youth employability using the ICT skills pyramid to explain the relevant industry skills as follows:

- Tier 1: This includes the basic industrial skills for employers such as team dynamics, problem-solving, and adaptability.
- Tier 2: This includes the various technical skills required in an ICT profession.
- Tier 3: This includes the unique industry type of abilities, knowledge, and skills that are more vulnerable to change [22].

The study developed a conceptual framework that described industries that targeted IT graduates with teamwork, social, and ICT technical skills (p. 21, [17]). This framework addressed the challenges that affected the improved employability of graduate youths in the labour market. The framework described three main components for future research as follows:

- The first component consists of graduate development.
- The second component consists of work experience in finding short-term work to enhance the graduates' experience.
- The third component consists of basic industrial hard and soft skills.

Therefore, it is important to realise that "youths attending educational institutions should have access to the relevant ICT skills required by employers in the labour market to improve their chances of employment" (p. 6, [22]). Furthermore, the industry skills required by youths are critical to succeed in competitive working environments. This is because employers value the basic skills and knowledge of ICT skills. Therefore, the training of ICT skills at educational institutions and grassroots using government-established ICT centres can prepare qualified youths to flourish in their career choices. This study shows that the

critical skills component needed in the labour market is soft skills as shown in the IT skill pyramid Tier 1 showing the necessary foundation skills (see Figure 1 above).

It is a reality that many employed youths struggle to communicate effectively with customers and co-workers due to a lack of practical skills (p. 1, [23]). However, qualified youths with relevant soft skills at the workplace tend to participate better in teamwork and communicate effectively [23]. Therefore, youngsters seeking employment opportunities require ICT skills that are equal to critical technical skills for effective teamwork and client engagement. We argue that basic computer knowledge and software application know-how are critical skills required in the labour market. Thus, the education sector needs effective collaboration with industries to improve the practical knowledge of learners to prepare them for the needs of the labour market.

### 2.2. Effects of Unemployment on Youth Development

The issue of youth unemployment in South Africa is caused by factors such as lack of ICT skills, access, income, affordability, infrastructure, poverty gap, inequality, lack of education, lack of information access, and high demand for IT skills expectations in organisations [14]. Other contributing factors that affect youths include stress caused by environmental and psychological factors that affect their health and mental stability [24]. The high unemployment rates in the country caused destructive behaviours among the youths causing mental health issues that have affected their ability to seek individual opportunities [24]. The lack of income generation has caused negative effects such as lack of confidence, suicide, stomach ulcers, substance abuse, and chronic behavioural conditions [24]. Other macro-social components include crime, divorce, and child abuse. The issue of unemployment affects the poor causing many people to lose faith in themselves and turn to destructive measures in South African society (p. 518, [24]).

A comprehensive report on the future of the South African workforce emphasised that the several challenges of unemployment faced in South Africa include the inequality existing in the country [14]. The World Bank advised that many countries need to channel resources towards educational platforms and organisations need to be ICT compliant to encourage their workforce to be ICT skilled and have increased competition [19], especially when digitisation can create opportunities for economic growth and contribute to individual development to reduce poverty [4]; this is lacking in the South African economy [19]. Studies have shown that the high unemployment rate in the country widened income inequality between the rich and the poor causing youth unemployment to increase drastically [15,25]. Many families live below an average income and cannot afford education for their offspring [15,26]. This has caused an underperforming education system that has caused many youths to become school dropouts and hindered their chances of receiving a quality education. While youths living in marginalized areas lack the opportunity to have a good education and livelihood, youths from affluent households have better opportunities to attend good schools and obtain good academic results from good-ranking educational institutions [14]. Although youths from different backgrounds obtained advanced education, some youths still lack the practical ICT skills required in the industry due to a lack of ICT access in their communities.

In addition, a lack of educational productivity contributes to limited jobs, especially as unemployment upsurges with reduced educational levels [14]. This has caused the education system not to generate the ICT skills required in organisations, and the labour supply is restricted by the growth of job seekers over time [14,27]. Many of these challenges are related to the lack of adequate ICT skill programs offered at educational institutions [14,27]. Therefore, practical ICT training that aligns with industry requirements should be incorporated into educational institution curricula at all levels [14]. This is because ICT skills can have multiplier effects across income levels and innovative capacity and can improve youth employability [3].

Other challenges include limited employment opportunities available to people who can afford private training courses or programs required in the labour market. There

is a need for policy changes that can compensate for youth ICT skills development by providing an agreement with the private sector to create employment opportunities for ICT-qualified youths. South Africa's infrastructure for Network Readiness ranked 65th out of 139 countries, and digital transformation is a crucial requirement in the country, especially as organisations are compelled to adapt to technological changes in their business strategies. This emphasises the existing skills gap in the labour market because many youths lack the relevant ICT, soft, hard, and technical skills required in industries for technological innovations [14]. In addition, many youths living in disadvantaged locations lack access to ICTs, such as mobile technology, due to a lack of income and information access to employment opportunities [15]. Thus, youths may have to gain the necessary ICT skills relevant to organisational needs to limit youth unemployment [17,27].

*2.3. Challenges in Labor Market Strategies for Employability*

Research has shown that youth unemployment in sub-Saharan African countries is caused by factors such as lack of investment, ineffective ICT skills, and uncomplimentary skills that the youth have in the labour market [28,29]. Research reports focused on youths from informal settlements in Uganda where an ICT initiative was implemented in the community to train life and entrepreneurial skills [29]. The project followed the Social Return on Investment Network (SROI) approach which comprised the economic, social, and environmental value created by an organisation [29]. It was a measurement tool that compared investments and values achieved and generated by the organisation [29]. The research concluded on the importance of ICT training for employability, the requirement of the labour market, and the link between industry and education institutions. Therefore, the implementation of internships and graduate programs can help graduates gain professional experience, improve their ICT skills, and build a professional network [5,21].

In addition, South Africa launched a Digital Industrial Revolution Commission in 2018, and the Department of Higher Education and Research Training Institutions conducted comprehensive research to investigate emerging technologies and the capabilities needed for advanced education using digital transformation to increase larger investments of cross-sector know-how for the development of entrepreneur and technical skills. The design of ICT educational programs can equip many graduates lacking ICT, soft and hard, technical, and business skills needed to seek employment. The study introduced Nano degrees to complement existing graduates, especially as innovation and digital transformation are imperative to enhance competitive advantage in the economy of many countries [30].

Furthermore, the study showed that graduates should be equipped with the necessary ICT skills that can provide employment opportunities in the labour market through the media, information applications, computer, and technological skills that will be used to solve problems, add value to employers and increase employment chances of qualified youths [30]. Hence, there is a continuous relationship between the change in technologies, employment opportunities, and the obtaining of ICT skills for improved youth employability through what youths are taught at educational institutions and ICT skills training acquired to prepare these youths for the labour market [30].

The research was conducted on the insufficiency of South African government strategies or initiatives for the ICT skills shortage in the country [30]. An option to budget for a new ICT university was suggested, but there was a lack of evidence to move the initiative forward [31]. This was to avoid unnecessary expenditures whenever there is no existing ICT skills shortage correctly identified. Therefore, existing gaps linking qualified ICT skilled youths and professionals employable in the labour market should be identified for policymakers to implement effective ICT training for different sector employees and potential employees such as youths at the grassroots level to prepare them for the labour market (p. 8, [3]).

### 2.4. Strategies to Promote Sustainable Youth Employment

Many organisations require more youths with postgraduate qualifications and ICT skills and educational institutions are expected to promote larger practical ICT skill-related courses in their pedagogy [2]. This is because employers find it challenging to hire qualified ICT youths during their employee recruitment process. Therefore, industries should develop marketing strategies to equip qualified ICT-skilled youths with demand in the labour market. In addition, there is a wide gap that affects the policies that mitigate youth unemployment in South Africa such as the minimum number of ICT-qualified youths that align with Black economic empowerment recruitment initiatives targeted at ICT-qualified youths and the adoption of an ICT curriculum in educational institutions to equip youths with valuable skills that will prepare them for the labour market.

Furthermore, the demand and supply of the labour market need to disseminate the necessary information about critical skills required for economic growth. Therefore, educational systems need to promote and incorporate entrepreneurial education and training in high schools, vocational schools, private colleges, and university curricula. Private colleges and universities should introduce mentorship programs to prospective youths through practical teaching methods using advanced digital tools that can introduce young learners to the demands of the labour market and guide the youths toward a fruitful career path (p. 73, [28]). Young professionals exposed to practical industrial training are equipped to perform better in the labour market. Furthermore, the intermediary connections in the labour market need to provide an integrated, solid approach to addressing unemployment issues through the implementation of policies to help employers target accurate skill levels.

## 3. Theoretical Framework: Sustainable Livelihood Theory

The relevance of ICTs in the framework embraces multiple dimensions interrelated in a unique way [11]. The Sustainable Livelihood framework attempts to link macro and micro factors such as structures and processes accessible to people using them as strategies to shape and constrain people's possibilities. The Sustainable Livelihood Theory process is wide and allows the use of social and cultural factors that shape livelihoods [11]. People work within factors such as the challenges they face, the assets they own or can access, the social reality of their culture, the system of government, and the private sector to weave livelihood strategies that result in livelihood outcomes. These outcomes can include increased income and savings, but more broadly improved well-being, reduced vulnerability to risk, improved future livelihood options, and the sustenance, or even replenishment, of natural resources. The aim is to help stakeholders with different perspectives and engage in structured and coherent debates about the many factors that affect livelihoods, their relative importance, and how they interact [11].

ICTs impact livelihood assets in several ways, depending on the local context in which they are introduced. The use of community public access models, digital centres or public access points such as the Acacia Initiative sponsored by the Canadian International Development Research Centre (IDRC) is an international effort to empower Sub-Saharan African communities with the competencies to incorporate information and communication technologies in the social and economic development of this African region, as well as "knowledge centres" such as the M. S. Swaminathan Research Foundation (MSSRF) initiative, a non-profit non-government organisation trust located in Chennai, India used to develop and promote strategies for economic growth to increase the employment of marginalized people in rural areas. Similar initiatives can be established to facilitate ICT skills training at the grassroots to motivate youths living in marginalized communities [32]. In the South African context, ICT initiatives such as Cape Access and Smart Cape in the Western Cape have provided access to ICT tools and infrastructures to people in poor areas [32]. Hence, public access points or ICT skills training centres can provide access to ICT tools and have a positive impact on the livelihood of youths.

### 3.1. Assets of Sustainable Livelihood Theory

The Sustainable Livelihoods approach is organised and categorised into five assets [11]. These five categories are as follows: Human Capital such as skills, knowledge, and ability to work or produce; Financial Capital such as funds available for investment, production, and consumption; Social Capital such as networks, participation in socially productive groups, and mutually beneficial relationships; Physical Capital such as buildings, infrastructure including power and water and productive tools, and Natural Capital such as natural resources. The relationship of these assets to the study is explained as follows:

- Human Capital: This involves access to ICT tools such as public access points (PAP) and public libraries for youths from disadvantaged communities [32]. The provision of digital centres in communities can provide youths with information about industry employment requirements, education and training, and educational tools in different formats. Access to ICT tools in disadvantaged communities can be significant to provide access to possible work opportunities and allow youths to make choices that can improve their livelihood [15].
- Financial Capital: This includes the strengthening of local financial institutions and micro-credit organisations to help youths gain access to useful information, financial services, and facilities on loans and savings schemes for improved livelihood [15].
- Social Capital: This refers to youths having improved "networking" both at the community level and with the people of authority, existing networks, and potentially among a much wider community. ICT tools are essential to youths because they can be used for professional networking in various contexts, including on social media platforms such as LinkedIn, to search for employment opportunities [15].
- Physical Capital: This refers to allowing youth entrepreneurs to gain improved access to market information and make strategic choices on the sale and purchase of goods from local markets which enhances the information on prices, comparative supply, and demand for products to sustain their business [15].
- Natural Capital: This includes improved access to organisations that can deal with different aspects of natural resource management and administrative and legal information. The provision of these resources will help youths gain access to the relevant resources that can be used to improve youth employability [15].

### 3.2. Development of Conceptual Model

The study developed a conceptual model to further explain the significance of ICTs to the sustainable development of youth employability. The study created a theoretical contribution to illustrate the impact of ICT on youth sustainability. The model explains the causes of youth unemployment as follows: lack of ICT skills, affordability, income, access, infrastructure, poverty gap, inequality, lack of education, lack of information access, and high demand for IT skill expectations in organisations. Further illustrations show that youth unemployability can be improved using the Sustainable Livelihood Theory Five Theory assets, namely human capital, financial capital, social capital physical capital, and natural capital, and their contribution to the personal development of youths. The theory was used to explain the prospective outcome of ICT skills training in society when practical training is integrated into educational institutions and government-sponsored computer training centres at the grassroots. This can prepare prospective ICT-skilled youths for industry skill demand and entrepreneurship skill opportunities. The model was presented to show the relevance of ICT skills for achieving the sustainable development of youth employability (see Figure 2).

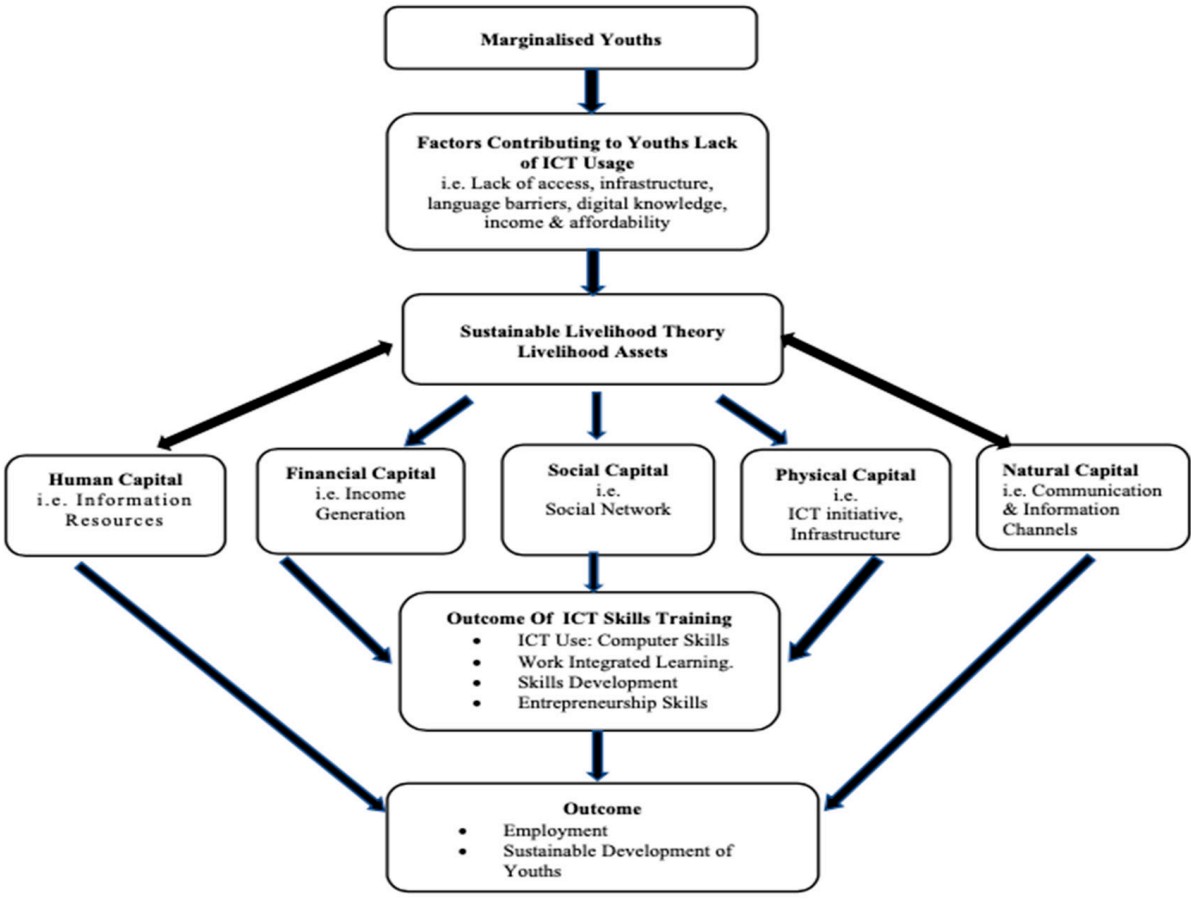

**Figure 2.** Conceptual model developed for the study.

## 4. Research Methodology

The researchers used a structured questionnaire survey that consisted of 29 closed-ended questions on various sections listed as follows: ICT and employability, quality of ICTs at educational institutions, barriers to youth access to ICT skills, the significance of ICT use for improved youth employability, ICT skills proficiency, ICT skills for information, and access to ICT resources at educational institutions. The researchers conducted a pilot study using a small sample size. For example, the study conducted a pilot study also known as a "feasibility" study to gather a small-scale population of respondents. The researchers conducted a preface study to test the groundwork of the fundamental stage of prospective research, which consists of a large-scale population of respondents. Hence, the preliminary research has been outlined to use quantitative research methods to evaluate the probable full-scale research project. The pretesting of the pilot study helped researchers identify problematic research questions that may lead to biased answers or be complicated for participants during the data collection process.

The structured questionnaire was distributed through various online channels or platforms such as Facebook, WhatsApp and LinkedIn, and a disclaimer mentioned in the survey was open to youth networks in Johannesburg. The quantitative research method used closed-ended, easy-to-answer questions that were direct, valuable, and descriptive to address the research problem investigated [33]. In addition, open-ended questions using quantitative questionnaire questions were adopted in the study to understand respondents' perceptions of the significance of ICT skills in educational systems for improved youth employability [33]. The study presents the profile of the respondents showing the respondents' gender and employment status (see Table 1).

**Table 1.** Demographics of respondents. Profile of respondents (N = 49).

| Ages | Female | Male | Total |
|---|---|---|---|
| 15–24 | 18 | 10 | 28 |
| 24–35 | 10 | 11 | 21 |
| Qualification | | | |
| Masters | 1 | 2 | 3 |
| Postgraduate | 6 | 10 | 16 |
| Degree | 2 | 2 | 4 |
| College | 4 | 4 | 8 |
| High School | 6 | 8 | 14 |
| Middle School | 2 | 2 | 4 |
| Employment | | | |
| Employed | 11 | 10 | 21 |
| Unemployed | 15 | 13 | 28 |

Additionally, the researchers did not incorporate advanced analysis such as confirmatory factor analysis in the study to determine the relationship between the factors influencing ICRT but rather triangulated the data derived from the study and conducted focused group discussions to allow the respondents to further express their responses. The researchers used quotes to describe the expressions received from respondents. The focused group discussion allowed respondents to provide an in-depth insight into their viewpoint about the importance of ICT skills for improved youth employability.

### 4.1. Data Collection

The researchers conducted data collection using random sampling to select the 49 participants involved in the data-gathering process [34]. Random sampling is a type of probability sampling in which participants are randomly selected from a subgroup of a population which is an unbiased representation of the total population [34]. The researchers conducted data collection using Google Forms to draft the questionnaire used for the data collection process. All data gathered were secured and protected using the SPSS software program through encrypted and password-controlled user accounts. Thereafter, data gathered were integrated into an Excel software file that was used to interpret information using graphical illustrations such as bar graphs and histograms to describe and explain the results of the data gathered in the study. Excel software program was used to recommend and create PivotTables to analyse, explore, and present the data gathered. This study used descriptive analysis to describe the characteristics of the sample of the data gathered from the study. The Sustainable Livelihood framework was used to guide the data gathered process. To triangulate the study, results derived from the respondent, answers were presented as quotes from the focus group discussions in the study. Youth Key Informant open-ended quantitative questions were used as a guide in the focus group discussions to understand the respondent's viewpoints about the importance of ICT skills on youth employability. The subsection highlights data analysis conducted in the subsection.

### 4.2. Data Analysis

The conceptual analysis was conducted using the Statistical Package for Social Science (SPSS) software program to analyse the data gathered. The conceptual analysis was conducted using a quantitative method to analyse the results. According to Swanepoel et al. (2010), statistical methods are a suitable way to summarise characteristics from data to reveal and interpret data's unknown traits gathered using tables, diagrams, charts, and numerical statistical measurements. The researchers ensured that all data gathered were secured and protected through a user account with password-controlled encryption.

Thereafter, the data gathered was integrated into an Excel file to interpret information using graphical illustrations such as bar graphs and histograms to compare and explain the results of the study. Excel was used to recommend and create PivotTables to analyse, explore, and present the data gathered. This study used descriptive analysis to explain the results. To measure the results derived from a target population, we gathered data and performed analysis using the theory of the study. To triangulate the study results, respondent answers were presented as quotes to support the study. Youth Key Informant open-ended questions were used to guide the focus group discussions to understand the respondent's viewpoints about the importance of ICT skills in educational systems. The demographics of respondents used in the study were analysed, and codes were used to represent the identities of the respondents involved in the focus group discussions. For example, code R1 was used to identify respondent 1, R2 identified respondent 2, etc.

## 5. Results

The results derived from the study show that many employed youths owned ICT devices which played an important role in helping youth search for information on various employment opportunities. The data shows the majority of respondents had access to mobile phones which were used to access the internet and possible employment advertisements. This result supports the importance of ICT skills for improved youth employability. Further results are highlighted in the subsections.

### 5.1. Respondents' Educational Level

There is a need to create ICT skills awareness in society to inform youths about the importance of ICT career goals [20]. The study developed a conceptual framework to show the significance of ICT skills to youth employability. This study shows respondents with educational qualifications are likely to familiarise themselves with technology devices that can be used to obtain ICT skills, especially when digitisation is the new norm for innovation and skills sustainability [4,6–11,13,16,17,20,22,23,27,34,35]. The study results show that 3 respondents have obtained a master's qualification, 16 respondents obtained a postgraduate qualification, 4 respondents obtained a degree qualification, 8 respondents obtained a college qualification, 14 respondents obtained high school qualifications, and 4 respondents obtained a middle school qualification.

Furthermore, the results show that respondents with post-graduate degrees were the most employed, while respondents with a middle school level of education were unemployed. Some respondents with bachelor's degrees and college certificates were unemployed because of the economic barriers that affect youth employment and the inability to afford further education and ICT skills training. In addition, study results partially agree that respondents with a high school education were significantly more often unemployed in South Africa without sufficient income [9]. The study shows the educational background of respondents (see Figure 3).

### 5.2. Human Capital: ICT Accessibility and Employability

Human capital refers to youths having access to ICT skills and knowledge to use ICT tools to access employment opportunities online. The results show 74% of respondents used ICT devices such as smartphones to access the internet frequently and 2% did not own ICT devices such as smartphones to access the internet to search for employment opportunities. This shows that many youths use ICT tools to assess job opportunities which is in line with the study of ResearchICTafrica.net [26]. The study shows respondents' frequency of ICT access (see Figure 4).

Do you use ICT devices to access the internet to access work opportunities?

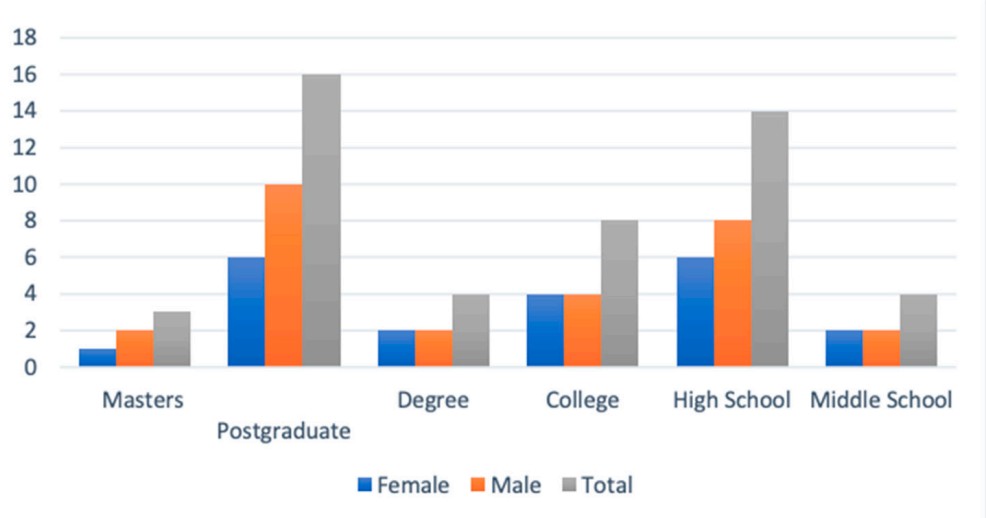

**Figure 3.** Respondents' educational background. Demographic question: What is your educational qualification?

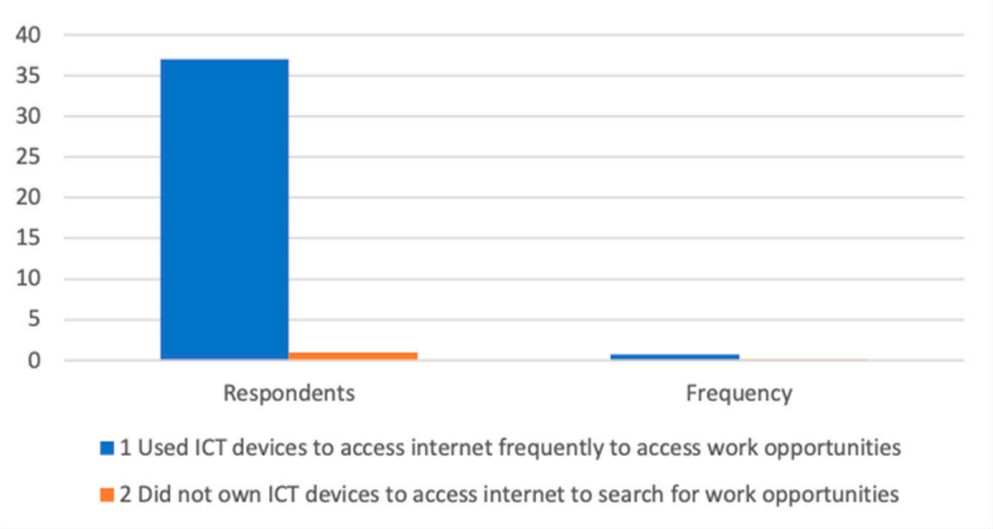

**Figure 4.** Youths' frequency of using ICT devices to search for employment through the internet.

The easiest and most convenient approach to seeking employment is through the use of ICTs. The researchers conducted a focus group discussion, and the following open-ended questions were asked: How frequently do you use ICTs to search for employment opportunities? Results derived from respondents' answers on the frequent use of ICT devices to access information about job opportunities state as follows:

> *"I use my mobile device every day to search on job sites available for work opportunities in the labour market".* [Focus Group-R1]

This study shows that the use of ICT devices has played an important role in helping respondents search for employment opportunities in the labour market.

*5.3. Natural Capital: Use of ICT Skills to Access Information and Gain Employment*

Youths should have access to information on various institutions that can provide the necessary natural resources which can be used to seek information on management, administration, legal, and other information that can be used to enhance their employment status using ICT skills [15]. The results suggest that 57% of respondents have gained employment due to their ICT skills, while 42% of respondents have not used their ICT

skills to gain employment. This study suggests the government should create strategies to provide access to ICT skills training to improve youth employability and bridge the digital divide [15]. Further results show that 89.9% of respondents used ICT skills to source information on industry requirements, 10% of respondents did not use their acquired ICT skills to source information on industry requirements for employment, while 98% of respondents used ICT skills to source information online, and 2% of respondents did not use ICT skills to source information online. The study results show that youths used ICT devices to seek information for employment purposes, industry information, and to seek information online. However, it is important to note that there are barriers that affect the use of ICTs for youth employability such as the lack of ICT skills, income, and infrastructure which is the cause of socio-economic problems [9]. The study shows the use of ICTs for industry information and employment opportunities (see Figure 5).

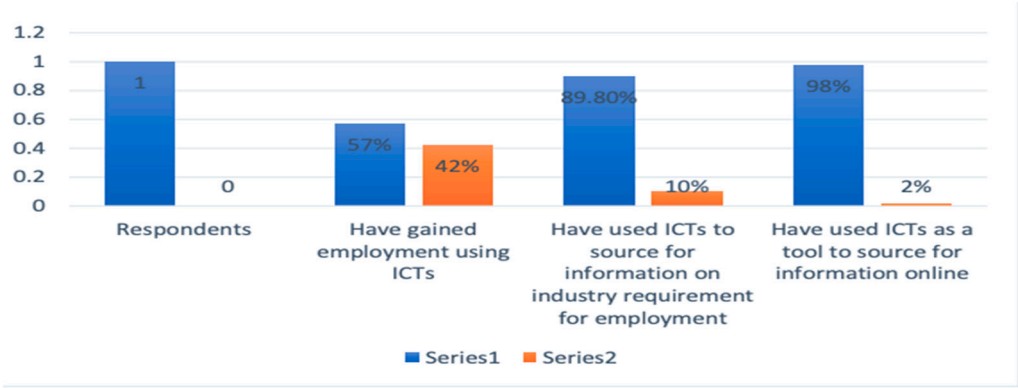

**Figure 5.** Respondents' frequency of access to ICT skills.

Respondents were asked the following question: Do you use ICTs to seek employment opportunities in organisations online? The results derived from respondents' answers on the use of ICTs to source information about industry requirements for prospective employment opportunities were claimed as follows:

*"I do use my mobile device to seek information about organisational job positions because some of my course mates now working in the industry try as interns, and I want to also get a position".* [Focus group-R3]

### 5.4. Financial Capital: Significance of ICT Use

The youths used ICTs to source information on opportunities from organisations that can provide the necessary services needed to improve their livelihood [15], for example, access to loans and savings schemes. The study results show that many youths were positive about using ICTs to improve their livelihood, and 26% of respondents used ICTs to increase their ICT skill knowledge and skill development. In addition, 32% of respondents used ICT devices to be aware of industry requirements for prospective employers, while 34% of respondents used ICTs to improve their skills and 34% of respondents claimed to use ICTs to be competent in the ICT skills required in the industry. Further results show 25% of respondents used ICTs for their competence in using different technological devices, while 30% of respondents used ICTs to attain their lifelong learning and 40% of respondents used ICTs to improve their economic standards. Study shows the significance of ICTs on youth employability (see Figure 6).

### 5.5. Physical Capital: Access to Quality ICT Tools

Youths should have access to quality ICT facilities to derive information on various opportunities that can help them make employment choices that will enhance their economic standards. Having the knowledge to use Microsoft software applications is a prerequisite for industry employment. Therefore, educational institutions should apply different theoretical and real-life practical use of ICT skills that youths can adopt to increase

their suitability for industry jobs. The results derived from the study show that 98% of respondents used Microsoft frequently, 89.80% of respondents used Excel frequently, 73.50% of respondents used PowerPoint frequently, 30.60% of respondents used Publisher Outlook and 42.90% of respondents used OneNote frequently. Many ICT tools are integrated into modern-day smartphones, while tablets are used for sales positions and management reporting. Many youths lack knowledge of using ICT tools such as fax machines which are least accessed by the respondents due to the perception that email use in the digital age is more acceptable than using a fax machine which is not mandatory.

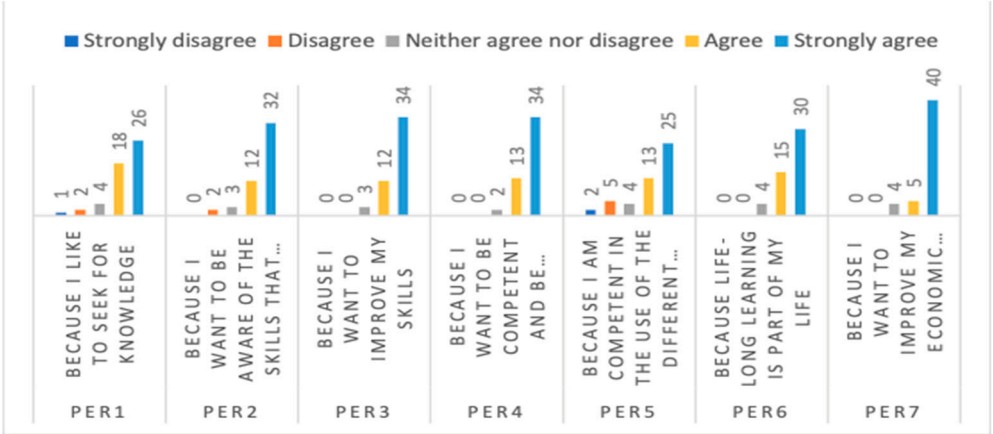

**Figure 6.** Significance of ICT use on respondents/ICT impact on knowledge, skills, and economic standards.

On the contrary, some organizations such as the government, health, and financial sectors still use fax machines in their daily work activities [36]. The use of slide projectors is rarely implemented, except for the exceptional breed of the device due to the affordability of the device which is mainly high. The study shows respondents' familiarity with soft applications and quality ICT tools (see Figure 7).

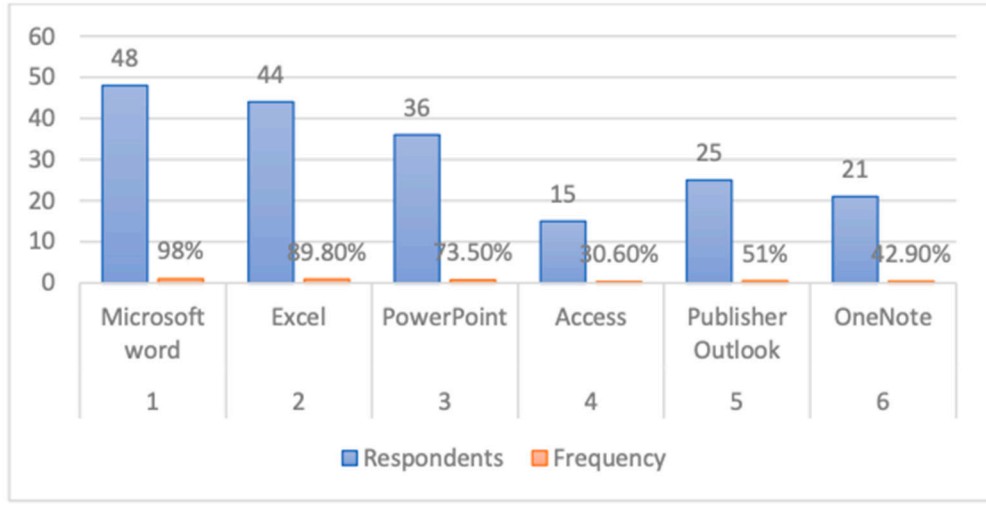

**Figure 7.** Respondents' use of quality ICT software: Microsoft package at educational institutions.

Furthermore, respondents were asked the following questions: How important do you think ICT skills are as a requirement in the labour market? Findings derived from the respondents show that ICT skills development is relevant for labour market demands and it has been realised that education systems need to incorporate ICT skills into their pedagogy as claimed:

*"As a youth, I need advanced digital skills knowledge before I can apply for industry jobs, also because many companies only employ people with Microsoft software application skills and other industry technology skills like soft and hard skills"*. [Focus group-R2]

*5.6. Social Capital: ICT as a Strategy for Youth Employment*

The easiest and most convenient approach to seeking employment was to use technological tools and it was determined that youths used ICT tools such as the internet for networking using social media networks [15]. The study results show respondents used social media platforms to search for employment. For example, advertisements on Facebook were used to showcase employment opportunities and other recruitment agency sites were used to search for employment opportunities [35]. The results show that respondents illustrate the importance of using social media such as LinkedIn to search for employment opportunities from organisations. LinkedIn provides the benefit of networking and the opportunity to seek employment with its easy-to-add digital professional profile which can be used to attract recruiters. Further results show the importance of the internet to search for employment opportunities, as respondents used the internet to seek employment on industry websites. The results show that 59.20% of respondents used LinkedIn to search for employment online, while 44.90% of respondents used Career Junctions, 12.20% of respondents used Glassdoor, and 36.70% used Indeed to search for employment online. Few respondents did not use the internet to find employment or have access to an appropriate smartphone or any ICT devices. The study shows the data pertaining to the use of ICTs to search for employment online (see Figure 8).

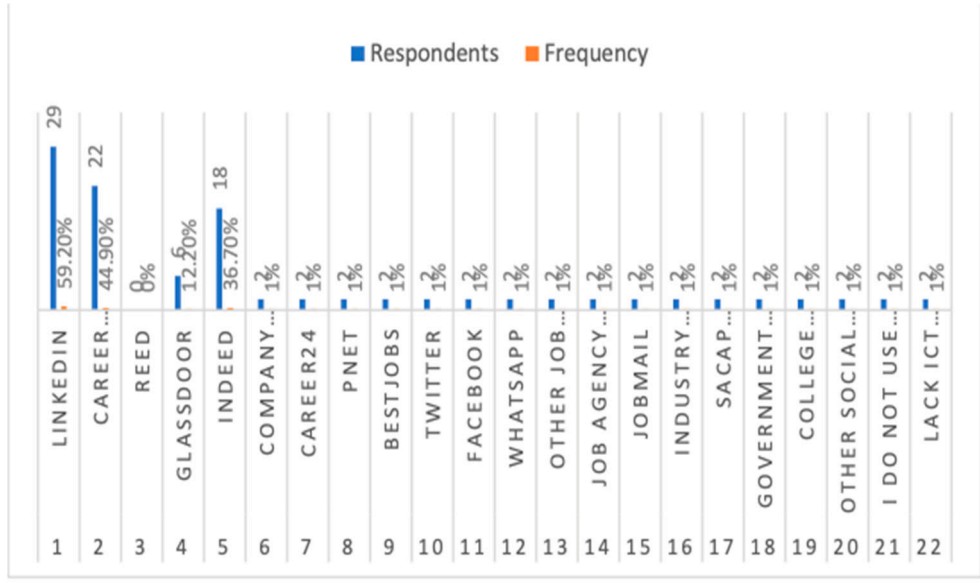

**Figure 8.** Employment measures adopted by respondents using ICTs to seek job opportunities in the labour market.

Additionally, respondents were asked the following question: Do you think the use of ICTs can enhance youth employment sustainability? The respondents attested that social media networks are a suitable medium to seek professional network and employment opportunities as claimed:

*"I mostly use my mobile devices to search online for professional work positions on LinkedIn and job adverts on recruitment sites because that is the only way I can get a job opportunity with my school certificate"*. [Focused group-R4]

*"Every day, I use my mobile device to check for job opportunities on social media sites. This is because sometimes online and remote jobs are advertised"*. [Focused group-R2]

### 5.7. Barriers to Adopting ICT Skills

This study compares to other studies that focused on the different factors that contribute to youth unemployment. It was determined that the causes of youth unemployment in the study were similar to other studies such as poverty gap, affordability, high demand for ICT skills, inequality, and lack of access are major challenges in South Africa [14,27]. Studies on ICTs interestingly suggest that youths in rural areas are more likely to use the internet than youths living in urban areas because of the availability of government-supported telecentres and other public access centres [37]. The results derived from the study show the major barriers to youths' ICT skills are as follows: lack of income, ICT skills knowledge, and government ICT training initiatives [37]. The study presents the results derived from (*n* = 49) respondents and shows that 83.7% of respondents lack livelihood, 51% of respondents lack knowledge, 61.2% of respondents lack computer skills, 57.1% of respondents lack government-sponsored ICT centres, 2% of respondents lack infrastructure, 2% of respondents lack income, 2% of respondents lack education and 2% of respondents lack access to information access (see Figure 9).

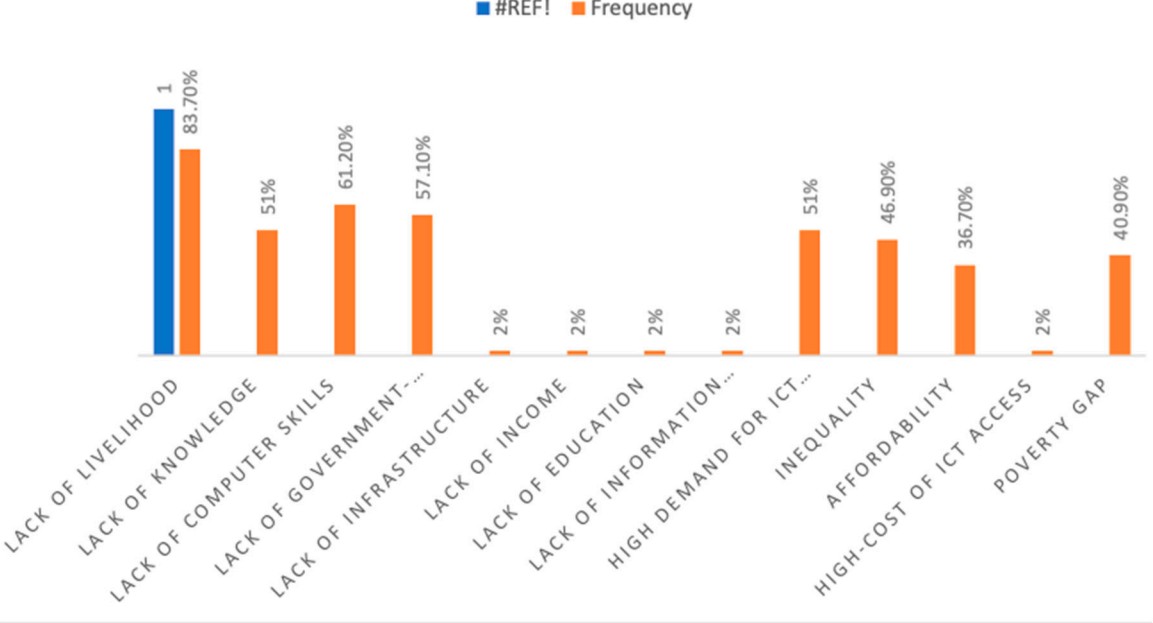

**Figure 9.** Respondent's reason for lack of access to ICT skills training.

Furthermore, respondents were asked the following question: What factors do you think affect your use of ICTs for information on improved economic standards?

The following barriers that contributed to youth unemployability were highlighted by respondents:

*"Many factors that affect youth unemployment in our country are the lack of ICT access, inequality among people, and poverty in our community because some people have education but are unemployed while some cannot afford education"*. [Focus group-R6]

*"Youth unemployment is causing poverty in many homes and many people because of this do not have access to ICT skills and ICTs to look for job opportunities"*. [Focus group-R5]

### 5.8. ICT Skill Relevance in the Education Systems for Improved Youth Employability

ICT skills are an important component relevant to the industry; therefore, educational institutions should adopt industry prerequisites into their curricula to equip youths for the labour market prospective employee demand. The study results show that 84% of the respondents claim ICT skills are relevant in educational institution curricula to prepare respondents for labour market demands and improve youth employability, and 16% of

respondents claim ICT skill preparation is not relevant in educational institution curricula to quip youths for labour market demands and improve youth employability. The results derived from this study further suggest that educational systems can adopt ICT skills as a suitable strategy to prepare youths for industry employment opportunities. Many youths claim ICTs are important for the sustainability of socioeconomic, information, and skills development to improve economic growth and cultural events. On the contrary, only a few respondents claim they were not convinced that ICT skills were important for youths to gain employment, but prefer to apply for jobs through networking, word of mouth, direct contact, and other traditional methods such as newspaper and billboard advertisements. The study shows respondents' perception of ICT skills for improved youth employability (see Figure 10).

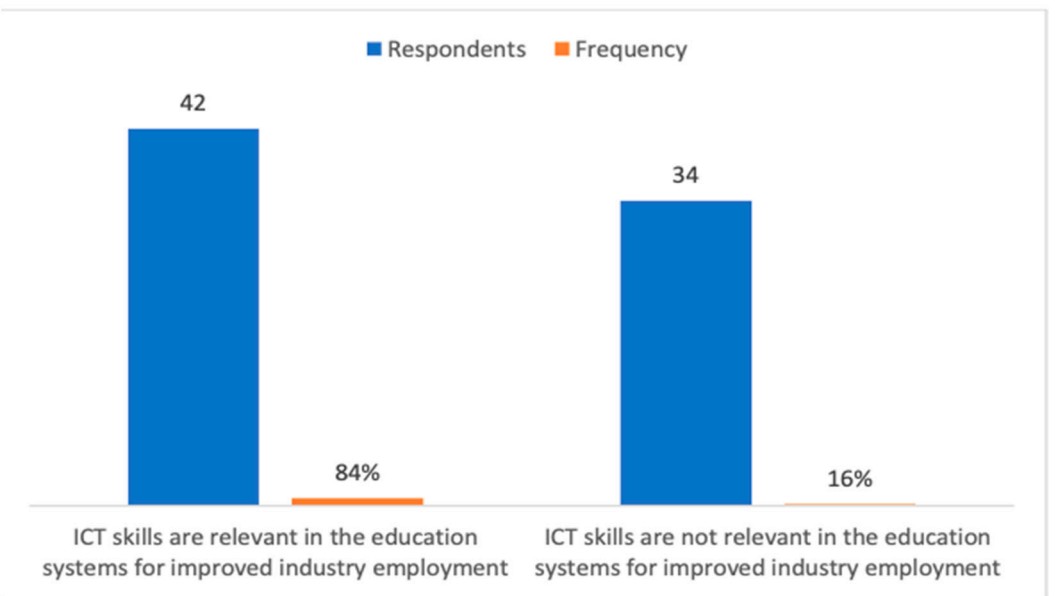

**Figure 10.** Response to the necessity of ICT skills for improved youth employability.

Respondents were also asked the following questions: What is your perception of ICT skills' relevance in the educational system to prepare youths for the labour market? Results derived from the study show that ICT skills are significant in the education systems for industry employment and to prepare youths for the labour market demand, especially as advanced digital skills are being promoted in many sectors such as educational, health, agricultural, mechanical, and agricultural industries due to the incorporation of the fourth industrial revolution in today's digital age. Therefore, educational systems are expected to adapt to the relevant industry skills demand that can assist youths to be equipped for the labour market requirement as claimed:

*"ICT skills are important for youths to gain employment in the industry, so the schools need to prepare young people for organizational employment".* [Focus group—R6]

### 6. Discussions

Technology is rapidly changing, and the digital age has become crucial in all aspects of humanity, including the industrial and education sectors. Hence, ICT skills are most relevant globally for economic growth, and youths are encouraged to acquire necessary ICT skills essential for the labour market. This is imperative in the South African economy due to economic barriers that prevent some youths from studying further due to their economic background hindering their ability to afford advanced education and inhibiting their ability to overcome the challenges of youth unemployment [9]. The results derived from this study have shown that policymakers should implement strategies for the early adoption of ICT skills at educational institutions. As many youths have high school and

higher education qualifications, this implies that educational institutions can incorporate ICT skills into their pedagogy [38]. Therefore, ICT skill learning in the classroom has a positive impact on both learners and educators, especially as many educators were trained through government initiatives. It would be imperative to adopt a similar learning trend in the education systems [39]. Therefore, youths with access to ICT devices can be channelled or trained to use advanced technologies required in the labour market.

Studies have shown that ICT skills have become imperative in the labour market, and educational systems should use the pedagogy that aligns with work-integrated learning to help youths gain practical industry experience. This study suggests that preparing youths to gain ICT skills and knowledge of other industry-relevant skills such as soft, hard, and technical skills will help youths meet the expectations of employees. Research from the study implied that many unemployed youths have obtained educational qualifications from high school to postgraduate. Only a few youths with limited employment obtained a master's degree, bachelor's degree, college certificate, and middle school qualification.

This shows educational institutions need to identify measures to equip qualified youths for organizational demands by incorporating practical learning modes using the work-integrated learning method into educational systems. In addition, youth employability can be addressed when educational institutions liaise with government, private organizations, and stakeholders to absorb educated youths as interns and vocational workers into their organizations to gain practical work experience. Results from the study show the gender disparity in youth employability. The results derived from this study show males were more employed than females.

*Theoretical Implication*

Furthermore, the study used the Sustainable Livelihood Theory as the theoretical framework used to guide the data-gathering process of the study [11]. An in-depth discussion of the theory focused on using five assets such as human capital, natural capital, financial capital, physical capital, and social capital to explain the impact of ICTs to improve youth livelihood [11]. The results derived from the human capital study show that youths with ICT skills can use the internet to access employment online more than youths without ICT access [11]. Hence, this study proposes that ICT skills training should be provided at both educational institutions and the grassroots level through the establishment of government-sponsored ICT skills training centres for youths living in marginalized communities.

The results obtained from the natural capital show that youths with ICT skills have access to information opportunities that have helped improve their livelihood. Therefore, this study suggests that the provision of ICTs to youths can help bridge the digital divide and improve youth employability in marginalized areas [11]. In addition, results from the financial capital show that youths with ICT skills could access information on possible opportunities such as loan access and savings schemes from government and private organizations to improve their livelihood [11]. Physical capital allows youths to access quality ICT facilities that can be used to access information on personal development and make employment choices that can improve the livelihood of themselves and their families. This shows the availability of ICT skills infrastructure in educational institutions is a suitable measure to train youths' ICT skills and improve their digital knowledge and entrepreneurship skills development.

Likewise, the availability of quality ICT facilities at the educational institution and grassroots levels can help youths access information on different career choices that will improve their livelihood [15]. Many youths are not familiar with the skills that are in demand in industries, while others cannot afford to own quality ICT devices such as iPad, laptops, smartphones, tablets, and ICT tools required for organizational sales transaction positions and management reporting. Furthermore, social capital allows youths to use ICT tools to access professional networks and connections that can improve their employment chances. This study proposes using ICT skills as a strategy to improve youth employment through the use of ICT internet to access social media networks and job professional

networks such as LinkedIn, Twitter, Instagram, and Facebook. This is because social media networks are used to advertise jobs, network opportunities with professionals, and for socializing purposes. Youths using these social media platforms mostly have an edge over youths without access to these networks due to the lack of ICT access.

Many organizational website links advertised online sometimes provide opportunities for youths to become online entrepreneurs using ICT skills. Although ICT tools such as smartphones are widely used for information and communication tasks, other ICT devices such as computers play a role in providing digital skills such as web design and administrative skills with the use of word processing software. The results from the study show that only a few youths were not interested in using ICT devices due to their lack of ICT skills, while many other youths used smartphones to seek employment opportunities.

## 7. Data Interpretation and Implication

The data interpretation of the study reveals the significance of adopting ICT skills in the educational systems for the sustainable development of youth employability in South Africa. In total, 84% of respondents claimed ICT skills were relevant in the educational systems to improve youth employability and 16% of respondents claimed ICT skills were not relevant to improve youth industry employment.

Furthermore, this study explored the factors that contribute to youth unemployment, and it was determined that the current economic drawbacks caused barriers that hindered youth employment in South Africa. This caused many youths' inability to afford ICT devices to access information on the relevant ICT skills required in the labour market. The study results from respondents revealed that 83.7% of respondents lack livelihood, 51% of respondents lack knowledge, 61.2% of respondents lack computer skills, 57.1% of respondents lack government-sponsored ICT centres, 2% of respondents lack infrastructure, 2% of respondents lack income, while 2% of respondents lack education and 2% of respondents lack access to information access.

In addition, the different factors that contribute to youth unemployment were caused by the poverty gap, affordability, high demand for ICT skills, inequality, and the lack of access which are major challenges that affected the South African economy [14,27]. Other challenges derived from the study results causing youth unemployment were the lack of income, ICT skills knowledge, and government ICT training initiatives. Additionally, the lack of livelihood contributed to the causes of low educational qualifications, as many youths could not afford to advance their education.

This study addressed the research problem and described the conditions in which ICT skills are a vital skills development component that needs to be integrated into the educational systems to overcome the existing high unemployment problem in South Africa. It has caused a lack of information literacy, ineffective communication within communities, and cultural activities, poverty, and inequality. In addition, studies have shown that poverty, inequality, lack of access, and high expectations of ICT skills from the labour market are major contributors to youth unemployment in this country [14,17,27].

Furthermore, ICT skills are a significant factor in the educational systems that embodies the relevant industry prerequisites in the education curriculum to equip youths for the labour market demands. This study reveals the importance of ICT skill educational systems to adopt ICT skills as an appropriate strategy to educate youths about industry demand for improved youth employability.

This study presents the potential impacts of ICT skills on youth employability and shows the increased demand for ICT specialists globally. In addition, the study discussed the proponents of the sustainable development goals (SDGs) such as SDG 4 target 4, which promotes the importance of relevant skills for decent employment; SDG 8, which promotes the need for sustained economic growth, inclusive and productive employment for all, and SDG 9, which is aimed at achieving greater access to technology for the improvement of livelihoods. Overall, the study contributed to the knowledge of the international academic environment focusing on information communication technology

for development (ICT4D) of global research that relates to policy-relevant and social science projects of the public sector users in sciences and human science research to address the aspect of using technology to tackle poverty caused by unemployment.

## 8. Practical Implications

This study suggests that policymakers and educational institutions should strategize ways to help youths gain ICT skills starting with Microsoft package software applications and implementing other software applications to use advanced digital technologies in the early stages of schooling. In addition, the educational systems and research training institutions should incorporate ICT skills such as software applications and the basic knowledge of innovative technology use, soft, hard, and technical skills into their pedagogy to develop the interest of graduate youths in innovative technologies [38,39].

The study further suggests that education systems should adopt advanced ICT skill strategies to improve the sustainable development of youth employability. Hence, the study recommends that policymakers propagate the awareness of ICT skills to improve youth employability, especially as digitization is a norm in the operation of organizations. Therefore, the government needs to execute strategies that support entrepreneurial skills training at educational institutions. The implementation of practical entrepreneurial education using advanced digital skills can be achieved through internships, vocational training, industrial project work, and real-life experiences of the labour market using work-integrated learning at educational institutions. This will help the educational system liaise with industries to absorb prospective qualified youths to undergo training that will prepare them for the labour market and help bridge the unemployment gap existing in South Africa.

ICTs can play an important role in the sustainability of skills development and improve youth employability in South Africa and other developing countries [28]. Youths can acquire ICT skills and knowledge to become young entrepreneurs because the world relies heavily on digital platforms, working from home with quality ICT tools and cloud services [20,31].

## 9. Study Limitation

The study limitations include the inability to gather more respondents at the study location due to the recent COVID-19 pandemic and the data collection process of the quantitative study. The study was cross-sectional and longitudinal, so the researchers were restricted from gathering a large number of respondents to participate in the study.

## 10. Conclusions

The study explored how ICT strategies can be used to improve youth employability in South Africa. The themes that emerged in the analysis were used to answer the research question of the study and indicated that ICT skills need to be incorporated into the educational systems for improved youth employability. The study used the Sustainable Livelihood Theory to provide an in-depth explanation of the five ICT asset categories, namely human capital, natural capital, financial capital, physical capital, and social capital, to promote the significance of ICTs on sustainable youth employability. The study explored the barriers hindering youth employability: lack of ICT access, income, affordability, infrastructure, education, and information access, as well as poverty gap and inequality, were determined to be the causes of youth unemployment. Furthermore, the lack of motivation from the education systems to channel youths to embrace the use of advanced technology skills is a major challenge that has contributed to the high unemployment rate. The study also explored the use of digital skills among youths to adapt to the fourth industrial revolution. In addition, future studies on the relevance of ICTs for economic empowerment and poverty reduction should be conducted to justify the results derived from this study. To justify the validity of the conceptual framework used for this study, a sequence of follow-up studies is recommended. Hence, future research can uncover the relationship between ICTs to sustain youth employability using ICT initiatives or programs used to reduce youth unemployment. Finally, further research should be conducted on the

implication of ICT skills on the gender divide and race discrimination that prevent youths from gaining sustainable employment.

**Author Contributions:** Conceptualization: A.A.; Methodology: A.A.; Software: A.A.; Validation: A.A. and R.B.; Formal analysis: A.A.; Investigation: A.A; Resources: AA and R.B.; Data curation: A.A; Writing original draft preparation: A.A.; Writing review and editing: A.A.; Visualization: A.A. and R.B.; Supervision: A.A. and R.B.; Project Administration: A.A. and R.B.; Funding acquisition: A.A. All authors have read and agreed to the published version of the manuscript.

**Funding:** This research was funded by National Research Foundation (NRF) South Africa grant number "138451" and "The APC was funded by the College of Business and Economics".

**Institutional Review Board Statement:** The study was conducted in accordance with the Declaration of the Applied Information Systems and approved by the Institutional Review Board (or Ethics Committee) of the University of Johannesburg (protocol code 2019_AIS_048 and 7/10/2019).

**Informed Consent Statement:** Informed consent was obtained from all subjects involved in the study. Written informed consent for publication was obtained from participants online publish this paper.

**Data Availability Statement:** The data that support the findings of this study are available in the article.

**Acknowledgments:** We acknowledge Govender Celina and Abosede Abubakre for their active participation in the research conducted.

**Conflicts of Interest:** The authors declare no conflict of interest. The funders had no role in the design of the study; in the collection, analyses, or interpretation of data; in the writing of the manuscript; or in the decision to publish the results.

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
