# Peer review of "Strategies for Using ICT Skills in Educational Systems for Sustainable Youth Employability in South Africa"

_sustainability, doi:10.3390/su142416513_

Round 1
Reviewer 1 Report
1.The article used a lot of content to explain that the use of ICT skills can promote youth sustainable employment, but it lacks detailed and specific suggestions and strategies to promote sustainable employment of youth. It can be said that the research problems, contents and methods do not match.
2.The relationship between ICT skills and digital skills was not clearly discussed. Is ICT skills equal to digital skills? For example, ICT skills were described in the abstract, but digital skills and ICT skills were used in the text.
3.The logic of the introduction part is confusing.
4.The first two paragraphs of the Study Context part are the same as the Introduction part.
5.Figure 1,3 are not clear.
6.The order of Human Capital, Social Capital, Natural Capital, Physical Capital and Financial Capital in lines 299-303 is inconsistent with that in lines 305-327.
7.What is the presentation of the Conceptual Model trying to illustrate? It is hardly used in the following paragraphs.
8.The Data Analysis section has some content (lines 356-358) that falls under the Data Collection section.
9.The poor presentation of Table 1 increased readers’ cognitive load.
10.The serial number of the figures is confusing, and it is also confusing when referring to the figure in the text (for example, figure 4 in line 413 is inconsistent with figure 4 below).
Author Response
Thank you for the suggested manuscript reviewer’s comment
“Please see the attachment” below.

Reviewer 2 Report
Dear Authors
Thank you for your efforts. I enjoyed reading the manuscript. In order to be suitable for publishing in this high impact journal, I suggest to improve your manuscript before making a decision to accept it. This includes improving the research problem statement and the contribution to the field on international and field levels.
Research design
I suggest to write more about your instrument, how do you develop it? validate? What is the context of your study?
Moreover, you data analysis should use advance statistical methods.
Please improve your discussion it is a superficial discussion. You missed the theoretical implications the the limitation of your study
Author Response

(The authors gave the same response as above.)

Reviewer 3 Report
The paper focuses on how higher education can adopt ICT strategies for the sustainable development of youth employability in South Africa. The methodology is a quantitative approach using a questionnaire. There were 49 out of 60 target respondents, youth people residing in East Rand, Johannesburg with different educational qualifications.
From the study emerges the significance of digitalization in youths and strong recommendations for higher education institutions to implement ICT strategies.
The paper looks interesting on a first glance, the topic is in line with the scope of the journal.
On the other hand, it looks like there is not so much care about the readability of the paper, with many little troubles.
Moreover, the questionnaire is quite simple in its nature (I do not know if there are other questions which were not considered in the graphical representation), and it is difficult to think that all what was promised in the abstract, the introduction and the background theory is detected by the questionnaire.
First two paragraphs of the discussion section look like an introduction. Then the conclusions are not derived from results (there should be a direct recall, otherwise it is not clear how you infer those conclusions).
Remarks
Line 19. in the abstract ”While the Sustainable Livelihood Theory was used to guide the study.” is strange, with no principal sentence, probably a typo somewhere. Same in Line 385.
Line 50: “among 15-24 years” → year olds
There are many double spaces
Figure 1 is of low quality, the text is not easily readable, it could be inserted in the text, there is no need to use a figure
line 175 is truncated
HEIs: once you state the acronym, it is not useful to entirely state it again (lines 184, 191, 192)
The order of figure 2 and 3 is switched
All the figures are low quality
Lines 399-400 recall figure 3 which is missing (it is not the figure 3 before)
Graphs could be better presented instead of just copying and pasting the images from google form
Figure 6 and Figure 9 are the same
Author Response
Thank you for the suggested manuscript reviewer’s comments
“Please see the attachment” below.

Reviewer 4 Report
Good theoretical background, but i would have liked a more solid statistical interpretation of data. I do not think you took advantage of all the data obtained, not all items were presented. You should discuss more about what the figures represent.
Also, check:
Line 18 -19 – wrong punctuation
Line 158 - youth unemployed – should be unemployment or rephrase
Lines 175-176 – should be in one paragraph
Line 382 - matric qualifications – please, explain the concept
Fig 2 – the levels of education should be ranked. From your fig I understand that middle school is higher than the masters’ degree
Author Response

(The authors gave the same response as above.)

Round 2
Reviewer 2 Report
Dear authors
Thank you for revising your manuscript based on my comments. You have addressed all my comments except advance analysis. You did a great job, but when I read the number of the participants which is low and data analysis was descriptive. Therefore, I suggest to use advance analysis such confirmatory factor analysis to study the relationship between the factors influencing ICRT or to use additional data collection for triangulation because in a big country we have only 49 who participated with superficial analysis it is difficult for duplication.
Author Response
Dear reviewer,
On behalf of my co-author and I, thank you for the constructive feedback given to our manuscript. We appreciate all suggested comments and have addressed them to the best of our ability.
Thanks
Kind regards

Reviewer 3 Report
There are still some elements that do not contribute to a global academic and professional soundness of the paper. For example in Figure 3 and 4 (which are improved compared to the previous versions) still contain some problems, for example the bars are called "series", while it should be stated what they mean (Yes, No,... or the question behind the agreement on certain topics). Probably a list of the questions or the presence of the related question close to the figure can benefit the readability of the paper.
Moreover, the sample size of the research is quite small, with 49 respondents one person represents more than 2% of the sample.
Discussion should connect with the results, otherwise the implication stated are not sufficiently corroborated.
I regret to suggest a reconsideration after revisions again, because I can see that the author made many efforts to adjust the paper, but anyway the scientific quality is doubtful
Author Response
Dear reviewer,
Thank you for the constructive feedback we appreciate your suggested comments and have addressed them all to the best of our ability.
Kind regards

Reviewer 4 Report
It is a much clearer presentation of your research now.
Author Response
Dear Reviewer,
We appreciate your constructive feedback.
Thank you.
Kind regards

Round 3
Reviewer 2 Report
Thank you
You have addressed all of my comments.
Author Response
Dear Editor in chief,
We the authors of the manuscript entitled: STRATEGIES FOR USING ICT SKILLS IN EDUCATIONAL SYSTEMS FOR SUSTAINABLE YOUTH EMPLOYABILITY IN SOUTH AFRICA.
thank you for the critical review and comments suggested to improve our manuscript standards. We appreciate all the reviewers' comments that helped to improve the quality of the research conducted and have responded to the comments highlighted.
Kindly find attached the updated manuscript and response to the reviewers' comments below.
On behalf of my co-authors, I apologize for the delay in the manuscript resubmission and appreciate your patience.
Thanks
Kind regards
Dr. Alao.
